# Interaction between Chocolate Polyphenols and Encapsulated Probiotics during In Vitro Digestion and Colonic Fermentation

Md Nur Hossain [1,2], Chaminda Senaka Ranadheera [1], Zhongxiang Fang [1] and Said Ajlouni [1,*]

1 School of Agriculture and Food, Faculty of Veterinary and Agricultural Sciences, The University of Melbourne, Melbourne, VIC 3010, Australia; mdnurh@student.unimelb.edu.au (M.N.H.); senaka.ranadheera@unimelb.edu.au (C.S.R.); zhongxiang.fang@unimelb.edu.au (Z.F.)

2 Institute of Food Science and Technology, Bangladesh Council of Scientific and Industrial Research, Dhaka 1205, Bangladesh

* Correspondence: said@unimelb.edu.au; Tel.: +61-3-83448620

**Abstract:** This study evaluated the interaction between probiotics and polyphenols in chocolates (45% and 70% cocoa) fortified with encapsulated probiotics. Cocoa powder was used as the main encapsulation component in a Na-alginate plus fructooligosaccharides formulation. Probiotic-chocolates (PCh) were produced by adding 1% encapsulated probiotics to the final mixture. The chocolate samples were subjected to in vitro gastrointestinal digestion and colonic fermentation. The data revealed that the most bioaccessible polyphenols in both formulations of PCh containing 45% and 70% cocoa were released in the gastric digested supernatant. The bioaccessible polyphenols from PCh with 70% cocoa reached 83.22–92.33% and 8.08–15.14% during gastrointestinal digestion and colonic fermentation, respectively. Furthermore, the polyphenols with higher bioaccessibility during colonic fermentation of both PChs developed with the CA1 formulation (cocoa powder 10%, Na-alginate 1% and fructooligosaccharides 2%) were detected in the presence of *Streptococcus thermophilus* and *Lactobacillus sanfranciscensis*. The results showed that PCh with specific probiotics favored the bioconversion of a specific polyphenol. For example, chocolate fortified with *Lacticaseibacillus casei* released larger quantities of epicatechin and procyanidin B1, while *Lactiplantibacillus plantarum* released more catechin and procyanidin B1 for *Lacticaseibacillus rhamnosus* LGG. Overall, the study findings concluded that chocolate polyphenols could be utilized by probiotics for their metabolism and modulating the gut, which improved the chocolates' functionality.

**Keywords:** chocolates; polyphenols; gastrointestinal digestion; gut microbiota; bioaccessibility

## 1. Introduction

There is a continuing interest in functional foods from both consumers and the scientific community [1]. These functional foods, also named as nutraceuticals, are known to provide additional health benefits beyond their nutritional values [2]. Some cocoa-derived products, such as chocolates, are very popular snack foods and can be grouped under functional snacks due to their high contents of flavonoids, a sub-group of polyphenolic compounds [3]. The flavonoids in chocolates include mostly flavanols, such as (−)-epicatechin (EC), (+)-catechin, and their dimers procyanidins B2 and B1 [4,5]. Chocolates also contain some other functional phenolic compounds, such as quercetin, iso-quercetin (quercetin 3-*O*-glucoside), hyperoside (quercetin 3-*O*-galactoside), quercetin 3-*O*-arabinose, luteolin, naringenin, apigenin, and methylxanthines, particularly, caffeine and theobromine [3,6]. Chocolates with these wide varieties of phytochemicals have been reported to exert beneficial preventive effects on the incidence and prevalence of many chronic diseases [4,7]. Bioavailability is determined by in vitro and in vivo methods using simulated gastrointestinal digestion, absorption and metabolism for determining the bioaccessibility of the bioactive compounds and nutrients through liberation from the food matrix, and assimilation by intestinal epithelium [8,9]. However, the bioavailability of flavonoids and other

phenolic compounds is crucial for achieving the potential beneficial impacts on human health [10]. In vitro analysis of the bioaccessibility of these phytochemicals provides meaningful insight into their bioavailability [11–13].

The consumption of live probiotics and their fortified functional foods can provide many health benefits for consumers, such as improved gut health, inflammation and allergies prevention, and immune maintenance [14–16]. Some recent studies have also reported that the chocolate flavonoids epicatechin (382 mg/100 g), catechin (115 mg/100 g), procyanidins (167 mg/100 g) and non-flavonoid compounds, such as theobromine (742 mg/100 g), are effective in modulating the gut microbiota composition [6,17], providing additional health benefits for the consumers. This could be due to the potential prebiotic effects of these compounds [18] that can lead to the production of specific fermentation by-products, such as short-chain fatty acids [19], bioactive amines, and amino acids [20] and B group vitamins [21]. However, the mechanisms of interaction between chocolate polyphenols and various probiotic bacteria in the gastrointestinal environment have not been fully investigated yet. This study aimed to investigate the bioaccessibility of chocolate polyphenols in the presence of some common probiotics (*L. rhamnosus* LGG, *L. casei* 431, *L. plantarum* UALp-05, *Lactobacillus acidophilus* La5, *Lactobacillus sanfranciscensis* JCM5668, *Bifidobacterium animalis* subsp. *lactis* BB12 and *Streptococcus thermophilus* UASt-09), using an in vitro gastrointestinal digestion and colonic fermentation model.

## 2. Materials and Methods

### 2.1. Chemicals and Reagents

The probiotics *L. rhamnosus* LGG, *L. casei* 431 (Lc) and *B. animalis* subsp. *lactis* BB12 cultures were gifted by Chr. Hansen, Melbourne, VIC, Australia, and *L. plantarum* UALp-05 (Lp), *L. acidophilus* La5, *L. sanfranciscensis* JCM5668 (Ls), and *S. thermophilus* UASt-09 (St) were obtained from the culture collection stock in the Food Microbiology Laboratory, School of Agriculture and Food, the University of Melbourne. Epicatechin (EC), catechin (C), procyanidin B1 (Pro B1), procyanidin B2 (Pro B2), quercetin 3-*O*-glucoside (QC glu), quercetin 3-*O*-galactoside (QC gal), Folin–Ciocalteu phenol reagent, gallic acid, HPLC grade acetonitrile, methanol, enzymes (salivary α-amylase, pancreatin, porcine pepsin), HCl, fructooligosaccharides (FOS) from chicory, degree of polymerization >10 and acetone were purchased from Sigma-Aldrich (Castle Hill, NSW, Australia). The selective culture media DeMan, Rogosa and Sharpe (MRS), yeast extract, protease peptone, beef extract, L-cysteine hydrochloride, bile salts, and trichloroacetic acid were bought from Thermo Fisher (Melbourne, VIC, Australia). Phosphate-buffered saline (PBS), potassium persulfate, potassium acetate, dextrose, aluminum chloride, NaOH, $(NH_4)_2SO_4$, $CaCl_2$, $K_2HPO_4$, $MgSO_4 \cdot 7H_2O$, NaCl, $NaHCO_3$, KCl, $MgCl_2(H_2O)_6$ and $(NH_4)_2CO_3$ were purchased from the Chem-Supply Pty Ltd. (Melbourne, VIC, Australia). Cocoa powder (100% pure) and other chocolate ingredients were purchased from a local supermarket (Melbourne, VIC, Australia).

### 2.2. Methods

#### 2.2.1. Preparation of Encapsulated Probiotics

The encapsulation of activated probiotics was conducted using the following two formulations: (1) cocoa powder 10%, Na-alginate 1%, and fructooligosaccharides (FOS) 2%; and (2) cocoa powder 10% and FOS 2% (without Na-alginate). These two formulations were designated as CA1 and CA2, respectively, and subjected to a freeze-drying technique according to the method described by Hossain et al. [22]. In this method, all probiotics in powder forms were revived and cultured anaerobically using an anaerobic incubator with 8% $CO_2$ (D-63450, Heraeus Instruments, Hanau, Germany), and cell pellets were harvested by centrifugation (Allegra X-12R, Beckman Coulter, Lane Cove, NSW, Australia). The pellets were washed and mixed with previously prepared encapsulation formulations CA1 and CA2. The formulation blends and probiotics were frozen overnight at −20 °C, before they were freeze-dried at −55 °C (Dynavac Engineering FD3, Seven Hills,

NSW, Australia). Chocolate treatments were prepared using 45% and 70% cocoa powder, and probiotics (encapsulated and non-encapsulated) were added at 1% ($w/w$) of the encapsulated probiotic powder. The prepared treatments included (chocolate fortified with encapsulated probiotics) a positive control (chocolate fortified with non-encapsulated probiotics) with individual probiotics and a negative control (basal medium and fecal slurry 5 mL each). All samples were investigated through in vitro gastrointestinal digestion and colonic fermentation in triplicates.

2.2.2. The In Vitro Gastrointestinal Digestion and Colonic Fermentation of Probiotic-Chocolate

The bioaccessibility of polyphenols was investigated using an in vitro gastrointestinal digestion and colonic fermentation model. The gastrointestinal digestion was conducted using salivary, gastric, and intestinal fluids according to the methods described by Minekus et al. [23]. The stock digestion fluids were made using a mixture of the electrolytes (K+, Na+, Cl, $HCO_3$, $H_2PO_4$, $NH^+$ $Mg^{2+}$, and $Ca^{2+}$) at different concentrations. The basal medium for the colonic fermentation was made following the methods of Hossain et al. [22]. A healthy male donor (32 years old), who had not ingested antibiotics or hormone supplements with any other complication for the last 3 months, donated fresh feces. The fecal slurry mixture was prepared as described by Tzounis et al. [24] and used on the same day of collection and preparation. The pH of the basal medium was adjusted to 7.0 using 1M HCl or 1M NaOH immediately before autoclaving. The in vitro colonic fermentation was investigated by mixing the gastrointestinal digested sample residue with the fecal slurry at a 1:1 ($v/v$) ratio, followed by anaerobic incubation at 37 °C for up to 72 h [22]. The in vitro gastrointestinal digestion was conducted in a 50 mL falcon tube incubated in an anaerobic incubator with 8% $CO_2$. The in vitro colonic fermentation was conducted in a fermentation flask and the anaerobic environment was created by flashing with $N_2$ gas. Sufficient replications were used to avoid any disturbance to the experiment. The samples for the total polyphenol and individual flavonoid analyses were collected at the end of 4 h of in vitro digestion and 72 h of colonic fermentation.

2.2.3. Extraction of Polyphenols from the Digested/Fermented Chocolate Samples

To avoid excessive fat in chocolates, the fresh chocolate samples (approximately 0.5 g each) were defatted 3 times successively with 5 mL of hexane and the residues were dried at 60 °C to evaporate the hexane completely [25]. The residues were extracted 3-fold with 2.5 mL of acetone: water: acetic acid (70:28:2, $v/v/v$) by sonication (20 kHz, 15 min) and centrifugation at $2500 \times g$ using a refrigerated centrifuge, and the supernatants were collected. The polyphenols were extracted from the supernatants after gastric digestion and colonic fermentation of all the treatments (samples, positive and negative controls). Each individually collected supernatant was concentrated to 2–3 mL by a vacuum evaporator (G3B, Hei-VAP, Schwabach, Germany) and diluted to 10 mL in Milli-Q water. All the diluted supernatants were filtered through a 0.22 μm membrane cartridge (Sigma Aldrich, Castle Hill, NSW, Australia) before the analysis of total polyphenols and individual flavonoids.

Analysis of Total Polyphenols Content in the Extracted Supernatant

The total polyphenol content (TPCs) in the extracted supernatants collected after gastric digestion and colonic fermentation was quantitated using the Folin–Ciocalteu method [26]. An aliquot of the extracted supernatant (100 μL) was incorporated with 100 μL of Folin–Ciocalteu's phenol reagent in a test tube and kept to react at room temperature for 10 min. Then, 300 μL of 20% ($w/w$) sodium carbonate solution and 1000 μL of Milli-Q water were mixed into the test tube, mixed thoroughly, and incubated at 40 °C for 30 min. The absorbance was measured at 765 nm using a microplate reader (Multiskan GO, Thermo Scientific, Melbourne, VIC, Australia). A standard curve was created using gallic acid and the results were calculated and reported as mg/g (GAE).

Analyses of Individual Flavonoids Using an HPLC Technique

The identification and quantitation of individual flavonoids in the extracted supernatants were performed by a Waters 2690 Alliance HPLC machine, equipped with a Waters 2998 photodiode array (PDA) detector (Waters, Rydalmere, NSW, Australia). A Gemini C18 Silica 250 × 4.6 mm, 5 μm column was employed with a binary gradient system (Phenomenex, Lane Cove, NSW, Australia). The filtered extracts (20 μL) were injected into the system with a binary phase: phase A (Milli Q water and 0.1% formic acid); phase B (acetonitrile and 0.1% formic acid) with a flow rate of 0.3 mL/min and 55 min gradient elution at a wavelength of 280 nm [27]. A standard curve was generated using six standard flavonoids (epicatechin, catechin, procyanidin B1, procyanidin B2, quercetin 3-*O*-galactoside, quercetin 3-*O*-glucoside) that were previously identified in cocoa powder and chocolate [28]. The detected individual flavonoids were quantitated using the matching peak area in the external standard along with the generated linear regression equation and expressed as μg/g of probiotic-chocolate. The instrumental limit of detection (LOD) and limit of quantification (LOQ) for the Waters 2690 Alliance separation module with a Waters 2998 PDA detector were calculated for each of the flavonoids. The LODs and LOQs for all the 6 polyphenol compounds were 2.0 μg/mL and 6.0 μg/mL, respectively.

*2.3. Statistical Analysis*

All the experiments were investigated in triplicate, with at least two measurements of each parameter. The results were subjected to one-way ANOVA using Minitab®19 statistical software (State College, PA, USA). The averages were separated using Tukey honest significant difference (HSD) at a 95% confidence level. The results were expressed as mean ± standard deviation.

**3. Results**

The encapsulation efficacy of probiotics using CA1 and CA2 formulations during processing and storage at 4 °C and 25 °C for up to 180 days, and during the gastrointestinal digestion were comprehensively described in our previous publication [22]. In the current investigation, the interactions between seven encapsulated strains of diverse probiotics and chocolate polyphenols were thoroughly examined using in-vitro gastrointestinal digestion and colonic fermentation.

*3.1. Total Polyphenol Bioaccessibility in Chocolate Fortified with Encapsulated or Non-Encapsulated Probiotics during the In Vitro Gastrointestinal Digestion and Colonic Fermentation*

The TPCs in the negative control (basal medium and fecal slurry), positive control (chocolate fortified with non-encapsulated probiotics), and probiotic-chocolate (chocolate fortified with encapsulated probiotics) samples containing 70% and 45% cocoa powder were determined before and after gastrointestinal digestion and colonic fermentation (Figure 1). The chocolate positive control samples showed that the average TPCs before in vitro digestion was 4.51 ± 0.18 and 2.81 ± 0.05 mg GAE/g in 70% and 45% cocoa powder, respectively, for all 7 tested probiotics (LGG, LC, La5, Bb12, Lp, Ls, and St) (Figure 1A). The data in Figure 1A also revealed that the total amount of bioaccessible TPC after the gastrointestinal digestion of the positive control chocolate was 87.34–92.33% and 88.72–95.61% in 70% and 45% cocoa chocolates, respectively. However, significantly ($p < 0.05$) less bioaccessible total polyphenols (21.06–26.60%) and (14.57–20.19%) were recorded after the colonic fermentation of the 70% and 45% cocoa chocolate positive control.

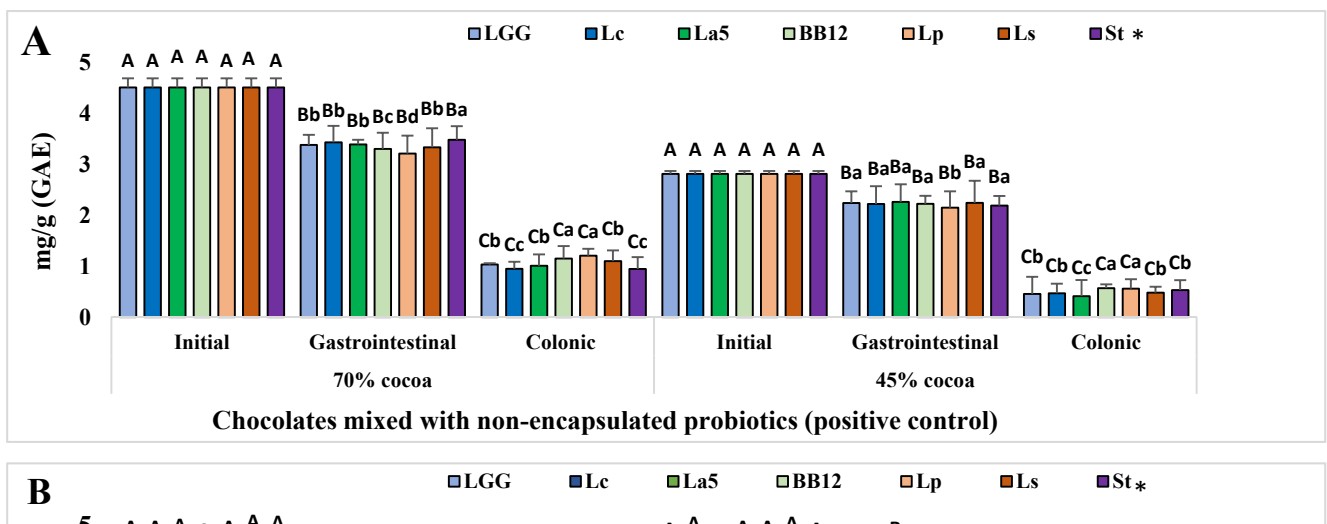

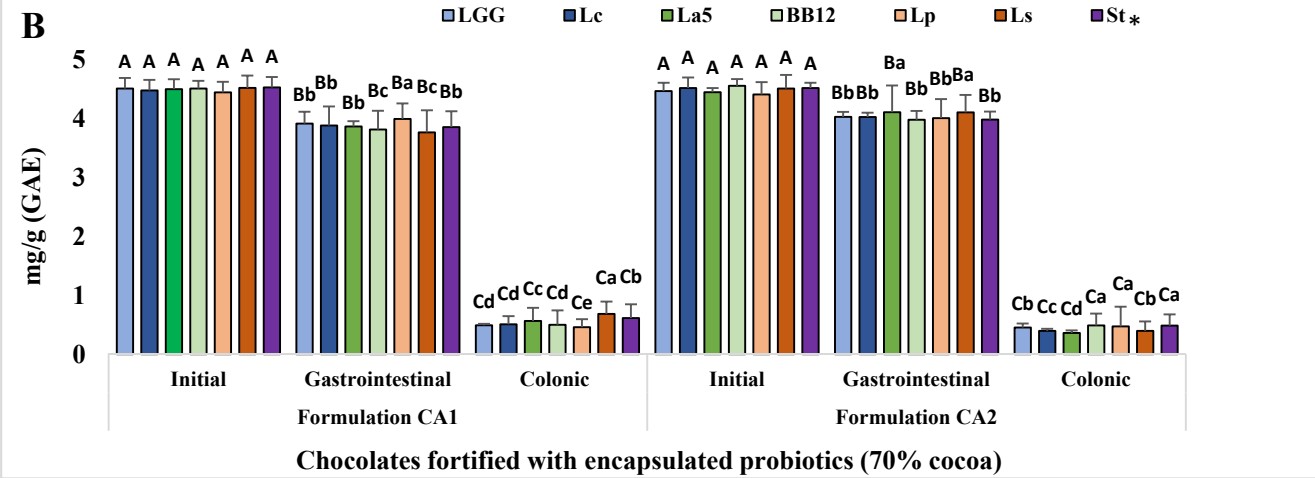

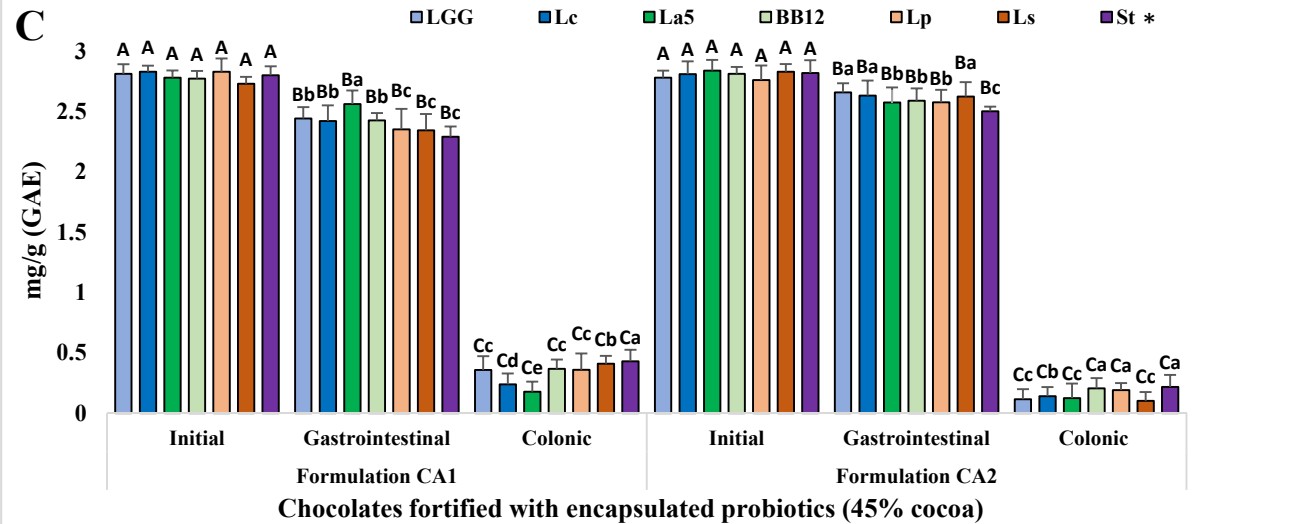

**Figure 1.** Total polyphenol content in probiotic-chocolate subjected to gastric digestion and colonic fermentation. (**A**) positive control, (**B**) probiotic-chocolate with 70% cocoa mass, and (**C**) probiotic-chocolate with 45% cocoa mass. * BB12: *B. animalis* spp. *lactis*, La5: *L. acidophilus*, Lc: *L. casei*, LGG: *L. rhamnosus*, Lp: *L. plantarum*, Ls: *L. sanfranciscensis*, St: *S. thermophilus*, *n* = 6. Columns with different uppercase superscript letters in different treatments indicate statistically significant (*p* < 0.05). Column with different lowercase superscript letters within each treatment indicate statistically significant (*p* < 0.05).

Similar to the results obtained with the positive controls (Figure 1A), the data from the probiotic-chocolates with 70% cocoa powder (Figure 1B) showed that the most bioaccessible polyphenols (83.22–92.33%) were available during the in vitro gastrointestinal digestion. In comparison, significantly ($p < 0.05$) smaller amounts (8.08–15.14%) of total polyphenols were available during the invitro colonic fermentation. The larger and smaller polyphenol quantities retrieved from the chocolate containing 45% cocoa powder with the CA1 formulation and *L. acidophilus* La5 and *S. thermophilus* after in vitro gastrointestinal digestion were 2.56 ± 0.11 mg/g and 2.29 ± 0.08 mg/g, respectively (Figure 1C). In comparison, these values reached 3.87 ± 0.09 mg/g and 3.86 ± 0.27 mg/g in chocolate with 70% cocoa powder under the same conditions (*L. acidophilus* and *S. thermophilus* after in vitro gastrointestinal digestion). As explained earlier, such a variation could be attributed to the greater contents of cocoa powder in the 70% probiotic-chocolates.

The interaction between the individually tested probiotics and chocolate flavonoids was analyzed. The polyphenol bioaccessibility showed no significant differences ($p > 0.05$) within each treatment (undigested, gastrointestinal, and colonic) in chocolates with 45% and 70% cocoa powder, with the following minor exceptions. Chocolates containing 70% cocoa powder with the CA1 formulation (Figure 1B) exhibited significantly ($p < 0.05$) larger amounts of bioaccessible polyphenols in the presence of *L. plantarum* (3.99 ± 0.26 mg/g) after gastrointestinal digestion. Furthermore, *L. sanfranciscensis* and *B. animalis* revealed smaller contents (3.77 ± 0.37 mg/g and 3.82 ± 0.31 mg/g, respectively) under the same conditions. An opposite trend was observed after colonic fermentation with the smallest bioaccessible polyphenols (0.46 ± 0.13 mg/g and 0.36 ± 0.04 mg/g, respectively) detected in chocolate fortified with *L. plantarum* and La5 in both CA1 formulations and 70% cocoa. The data revealed that 97–99% and 80–92% of total polyphenols were recovered from the probiotic-chocolates with 45% and 70% cocoa powder, respectively, after in vitro gastrointestinal digestion, and the remaining insignificant amounts were detected after colonic fermentation.

### 3.2. In Vitro Bioaccessibility of Individual Flavonoids from the Probiotic-Chocolates

3.2.1. Probiotic-Chocolate Containing 70% Cocoa Powder

The analyses of individual flavonoids in probiotic-chocolates (PCh) in both CA1 and CA2 formulations and positive controls after in vitro gastrointestinal digestion and colonic fermentation showed equivalent results. Furthermore, the negative control samples did not show any detectable peaks against the standard flavonoids. Consequently, the data in Tables 1 and 2 presented the results generated from the CA1 formulation only before and after in vitro gastrointestinal digestion and colonic fermentation. The most identified and quantified flavonoids in chocolates containing 70% and 45% cocoa powder included EC, C, Pro B1, Pro B2, QC gal and QC glu. The EC contents in the PCh fortified with *L. rhamnosus* LGG were 298.66 ± 5.2, 183.0 ± 1.59, and 87.33 ± 2.16 µg/g, before digestion, and after in vitro gastrointestinal digestion and colonic fermentation, respectively (Table 1). The values indicated that about 61.27% of EC was bioaccessible in the supernatant of the in vitro gastrointestinal digestion and 29.24% after the colonic fermentation. The +(−) catechin appeared with the smallest quantity (114.89 ± 3.16 µg/g) in the undigested PCh fortified with BB12, which yielded 75.33 ± 1.8 and 30 ± 1.52 µg/g after in vitro gastrointestinal digestion and colonic fermentation, respectively. Another two major flavonoids found in the chocolates were pro B1 and pro B2. The pro B1 contents in PCh fortified with *L. acidophilus* La5 were 388.46 ± 1.15, 291.33 ± 3.56 and 60.83 ± 2.19 µg/g initially, after in vitro gastrointestinal digestion and after colonic fermentation, respectively. The pro B2 contents in PCh fortified with *L. plantarum* were much higher than all the other detected flavonoids and showed 793.56 ± 3.98, 553.33 ± 2.24, and 191.33 ± 4.7 µg/g for undigested, after in vitro gastrointestinal digestion and colonic fermentation, respectively.

**Table 1.** Quantitates (µg/g) of major flavonoids isolated during in vitro digestion of probiotic-chocolate (70% cocoa powder).

| Probiotic-Chocolates | Digestion Stage | Flavonoids, µg/g | | | | | |
|---|---|---|---|---|---|---|---|
| | | EC * | C | Pro B1 | Pro B2 | QC Gal | QC Glu |
| LGG *** | Initial | 298.66 ± 5.2 a **** | 115.66 ± 1.56 a | 387.86 ± 1.05 a | 793.0 ± 5.52 a | 644.33 ± 4.55 a | 587.0 ± 3.93 a |
| | GD | 1830.0 ± 1.59 b | 82.0 ± 1.22 b | 313.5 ± 1.60 b | 636.33 ± 5.07 b | 631.67 ± 3.50 a | 582.0 ± 1.28 a |
| | CF | 87.33 ± 2.16 c | 26.0 ± 1.3 c | 68.7 ± 0.12 c | 146.33 ± 1.33 c | nd ** | nd |
| Lc | Initial | 296.36 ± 4.12 a | 121.16 ± 2.16 a | 382.18 ± 2.23 a | 798.13 ± 3.22 a | 649.13 ± 2.51 a | 586.06 ± 4.06 a |
| | GD | 194.66 ± 2.06 b | 73.33 ± 1.08 b | 321.33 ± 2.92 b | 704.24 ± 1.68 b | 623.37 ± 2.59 a | 581.33 ± 4.81 a |
| | CF | 96.33 ± 2.82 c | 32.0 ± 2.0 c | 58.3 ± 0.2 c | 68.0 ± 2.24 c | nd | nd |
| La5 | Initial | 299.61 ± 1.27 a | 117.34 ± 1.06 a | 388.46 ± 1.15 a | 794.24 ± 2.52 a | 644.87 ± 3.76 a | 589.52 ± 6.09 a |
| | GD | 184.67 ± 1.58 b | 77.0 ± 0.66 b | 291.33 ± 3.56 b | 652.0 ± 4.29 b | 627.33 ± 6.53 a | 571.33 ± 3.24 a |
| | CF | 97.0 ± 2.13 c | 34.67 ± 0.05 c | 60.83 ± 2.19 c | 103.67 ± 2.03 c | nd | nd |
| Lp | Initial | 298.49 ± 7.09 a | 115.27 ± 2.09 a | 390.66 ± 2.05 a | 793.56 ± 3.98 a | 643.45 ± 1.79 a | 582.74 ± 7.81 a |
| | GD | 178.25 ± 2.15 b | 69.0 ± 1.0 b | 277.7 ± 3.79 b | 553.33 ± 2.24 b | 623.67 ± 4.86 a | 574.67 ± 7.0 a |
| | CF | 90.0 ± 0.71 c | 43.67 ± 1.5 c | 102.63 ± 2.58 c | 191.33 ± 4.7 c | nd | nd |
| Ls | Initial | 301.24 ± 6.02 a | 118.57 ± 1.02 a | 387.23 ± 1.67 a | 797.21 ± 3.05 a | 647.87 ± 3.03 a | 589.26 ± 6.41 a |
| | GD | 169.0 ± 2.80 b | 72.33 ± 1.3 b | 271.03 ± 2.64 b | 702 ± 3.77 b | 637.67 ± 2.5 a | 576.04 ± 3.66 a |
| | CF | 113.33 ± 1.83 c | 39.67 ± 0.93 c | 102.2 ± 2.54 c | 81.66 ± 1.13 c | nd | nd |
| BB12 | Initial | 298.52 ± 5.02 a | 114.89 ± 3.16 a | 389.56 ± 3.25 a | 793.83 ± 2.35 a | 644.73 ± 5.72 a | 587.49 ± 2.93 a |
| | GD | 192.33 ± 4.77 b | 75.33 ± 1.8 b | 304.7 ± 1.83 b | 719.33 ± 3.71 b | 638.36 ± 2.18 a | 581.33 ± 4.89 a |
| | CF | 96.67 ± 0.51 c | 30 ± 1.52 c | 74.97 ± 1.67 c | 69.23 ± 3.78 c | nd | nd |
| St | Initial | 293.26 ± 3.76 a | 119.52 ± 1.35 a | 388.11 ± 2.59 a | 791.53 ± 2.46 a | 645.74 ± 1.04 a | 588.55 ± 7.01 a |
| | GD | 164.12 ± 3.07 b | 88 ± 1.47 b | 266.36 ± 2.87 b | 632.0 ± 3.53 b | 623.67 ± 4.04 a | 564.67 ± 3.71 a |
| | CF | 123.33 ± 2.02 c | 25.67 ± 0.06 c | 112.03 ± 2.40 c | 134.37 ± 3.13 c | nd | nd |

* EC: (−)-epicatechin, C: (+)-catechin, Pro B1: procyanidin B1, Pro B2: procyanidin B2, QC gal: quercetin 3-*O*-galactoside, QC glu: quercetin 3-*O*-glucoside; GD: gastrointestinal digestion, CF: colonic fermentation ** nd: not detected, *n* = 6. *** BB12: *B. animalis* spp. *lactis*, La5: *L. acidophilus*, Lc: *L. casei*, LGG: *L. rhamnosus*, Lp: *L. plantarum*, Ls: *L. sanfranciscensis*, St: *S. thermophilus*. **** Means within each column and each treatment (initial, after gastric digestion, and after colonic fermentation) followed by the same superscript letters were not significantly different at a 95% confidence level.

**Table 2.** Quantitates (µg/g) of major flavonoids isolated during in vitro digestion of probiotic-chocolate (45% cocoa powder).

| Probiotic-Chocolates | Digestion Stage | Flavonoids, µg/g | | | | | |
|---|---|---|---|---|---|---|---|
| | | EC * | C | Pro B1 | Pro B2 | QC Gal | QC Glu |
| LGG *** | Initial | 187.0 ± 1.2 a **** | 75.62 ± 1.42 a | 257.13 ± 2.05 a | 518.33 ± 1.52 a | 410.0 ± 3.55 a | 379.0 ± 5.93 a |
| | GD | 110.0 ± 1.6 b | 60.6 ± 2.81 b | 198.67 ± 3.11 b | 458 ± 9.07 b | 397.63 ± 3.50 a | 364.0 ± 1.28 a |
| | CF | 70.33 ± 0.55 c | 13.93 ± 0.11 c | 57.7 ± 1.12 c | 61.13 ± 1.33 c | nd ** | nd |
| Lc | Initial | 182.67 ± 3.09 a | 75.12 ± 1.02 a | 263.27 ± 2.85 a | 528.61 ± 7.52 a | 405.27 ± 7.09 a | 372.54 ± 5.23 a |
| | GD | 122.66 ± 1.09 b | 63.67 ± 0.08 b | 208.0 ± 2.08 b | 464.66 ± 4.68 b | 373.37 ± 2.59 a | 357.33 ± 4.81 a |
| | CF | 62.0 ± 0.80 c | 8.67 ± 0.3 c | 53.3 ± 0.28 c | 48.0 ± 2.24 c | nd | nd |
| La5 | Initial | 187.66 ± 3.98 a | 73.52 ± 2.26 a | 258.21 ± 3.05 a | 513.41 ± 1.92 a | 410.57 ± 3.05 a | 380.13 ± 8.07 a |
| | GD | 114.33 ± 2.48 b | 62.16 ± 0.57 b | 200.66 ± 2.56 b | 409 ± 6.08 b | 401.24 ± 1.53 a | 370.27 ± 3.24 a |
| | CF | 71.0 ± 2.51 c | 11.0 ± 1.7 c | 50.83 ± 0.19 c | 101.1 ± 2.03 c | nd | nd |
| Lp | Initial | 189.23 ± 2.03 a | 77.62 ± 1.15 a | 257.73 ± 5.82 a | 518.97 ± 6.02 a | 415.03 ± 5.87 a | 377.19 ± 3.78 a |
| | GD | 111.67 ± 2.68 b | 60.33 ± 1.0 b | 179.40 ± 1.79 b | 386.33 ± 3.72 b | 398.67 ± 4.86 a | 374.67 ± 2.0 a |
| | CF | 72.33 ± 2.06 c | 12.56 ± 2.5 c | 72.43 ± 1.58 c | 121.33 ± 4.7 c | nd | nd |
| Ls | Initial | 190.12 ± 4.08 a | 73.44 ± 3.42 a | 255.55 ± 4.09 a | 523.74 ± 6.97 a | 409.44 ± 2.07 a | 379.47 ± 6.09 a |
| | GD | 117.0 ± 2.51 b | 64.33 ± 1.3 b | 201.03 ± 2.94 b | 447.67 ± 5.22 b | 387.67 ± 8.5 a | 366.04 ± 1.66 a |
| | CF | 59.33 ± 1.56 c | 8.23 ± 1.7 c | 43.2 ± 1.54 c | 61.66 ± 3.13 c | nd | nd |
| BB12 | Initial | 186.87 ± 1.02 a | 78.23 ± 1.56 a | 258.35 ± 5.27 a | 522.36 ± 3.52 a | 412.67 ± 7.28 a | 378.93 ± 4.34 a |
| | GD | 119.66 ± 1.6 b | 67.67 ± 1.6 b | 208.46 ± 4.83 b | 452.33 ± 5.05 b | 378.36 ± 2.18 a | 357.33 ± 4.89 a |
| | CF | 65.33 ± 3.8 c | 6.4 ± 1.73 c | 44.97 ± 2.67 c | 59.23 ± 3.78 c | nd | nd |
| St | Initial | 187.74 ± 2.12 a | 76.56 ± 3.78 a | 264.70 ± 4.85 a | 518.43 ± 6.28 a | 410.71 ± 4.05 a | 381.33 ± 5.14 a |
| | GD | 121.66 ± 2.21 b | 66 ± 0.06 b | 161.16 ± 1.28 b | 364.67 ± 1.53 b | 403.67 ± 6.04 a | 361.61 ± 2.83 a |
| | CF | 61.67 ± 2.02 c | 8.66 ± 0.09 c | 83.8 ± 1.40 c | 142.0 ± 3.13 c | nd | nd |

* EC: (−)-epicatechin, C: (+)-catechin, Pro B1: procyanidin B1, Pro B2: procyanidin B2, QC gal: quercetin 3-*O*-galactoside, QC glu: quercetin 3-*O*-glucoside; GD: gastrointestinal digestion, CF: colonic fermentation ** nd: not detected, *n* = 6. *** BB12: *B. animalis* spp. *lactis*, La5: *L. acidophilus*, Lc: *L. casei*, LGG: *L. rhamnosus*, Lp: *L. plantarum*, Ls: *L. sanfranciscensis*, St: *S. thermophilus*. **** Means within each column and each treatment (initial, after gastric digestion, and after colonic fermentation) followed by the same superscript letters were not significantly different at a 95% confidence level.

### 3.2.2. Probiotic-Chocolate Containing 45% Cocoa Powder

The individual flavonoid content in probiotic-chocolates (PCh) containing 45% cocoa was presented in Table 2. As with the PCh fortified with 70% cocoa powder, the 45% cocoa containing samples showed a similar trend. The highest bioaccessible epicatechin was found at 67.14% for *L. casei* and 38.22% for *L. plantarum* after gastrointestinal digestion and colonic fermentation, respectively. Similar to the observations reported in the PCh containing 70% cocoa powder, the quantity of detected catechin was the smallest among all the tested flavonoids. Catechin content ranged from 73.44 ± 3.42 µg/g in *L. sanfranciscensis* to the highest 78.23 ± 1.5 µg/g in BB12 in the undigested PCh (Table 2). Additionally, the quantities of Pro B2 in the PCh containing 45% cocoa were the highest within each tested probiotic after in vitro gastrointestinal digestion and colonic fermentation, followed by QC gal, QC glu, Pro B1, C, and EC.

### 3.3. Interaction between Flavonoids and Encapsulated Probiotics in the Chocolates

The interactive main effect plot (Figure 2A) demonstrated the interaction between the encapsulated probiotics in chocolates and the flavonoid production in the PCh containing 70% cocoa powder, with the total released flavonoids during the in vitro gastrointestinal digestion and colonic fermentation. The effect of individual probiotics revealed that *L. sanfranciscensis* had a better bioconversion effect on most identified flavonoids than other probiotics, except catechin. This observation was confirmed by the larger amount of epicatechin, procyanidin B1 and procyanidin B2 detected in the presence of this probiotic during the in vitro gastrointestinal digestion and colonic fermentation. For *S. thermophilus*, the bioaccessible flavonoid quantities were also very good except for procyanidin B2, where it was the lowest. Other than these two probiotics, BB12, *L. acidophilus*, and *L. plantarum* also showed a higher bioconversion capacity for catechin. The results from the interaction assessment between probiotics and flavonoids in the chocolate containing 45% cocoa powder are shown in Figure 2B. The highest amount of EC was available with *L. casei* and *S. thermophilus*, whereas for catechin, the highest amount was in the presence of *L. plantarum*. For procyanidin B1 and procyanidin B2, *L. casei*, LGG, and La5 showed a better effect over the other probiotics. Overall, the data in Figure 2 demonstrated that probiotic-chocolate fortified with a specific probiotic favored the bioconversion of a specific flavonoid. For example, *L. casei* released the largest quantities of EC and procyanidin B1, while *L. plantarum* converted more catechin, and LGG more pro B2 (Figure 2B).

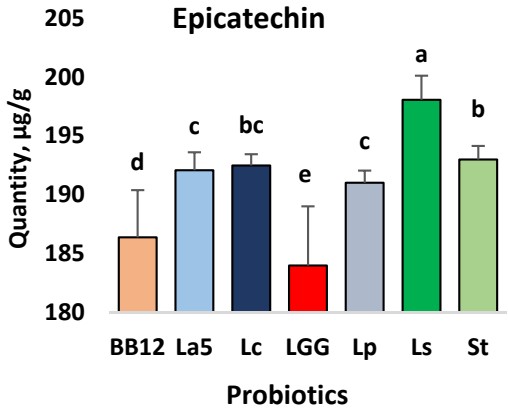
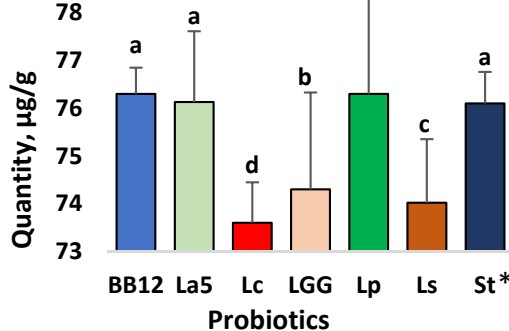

**Figure 2.** *Cont.*

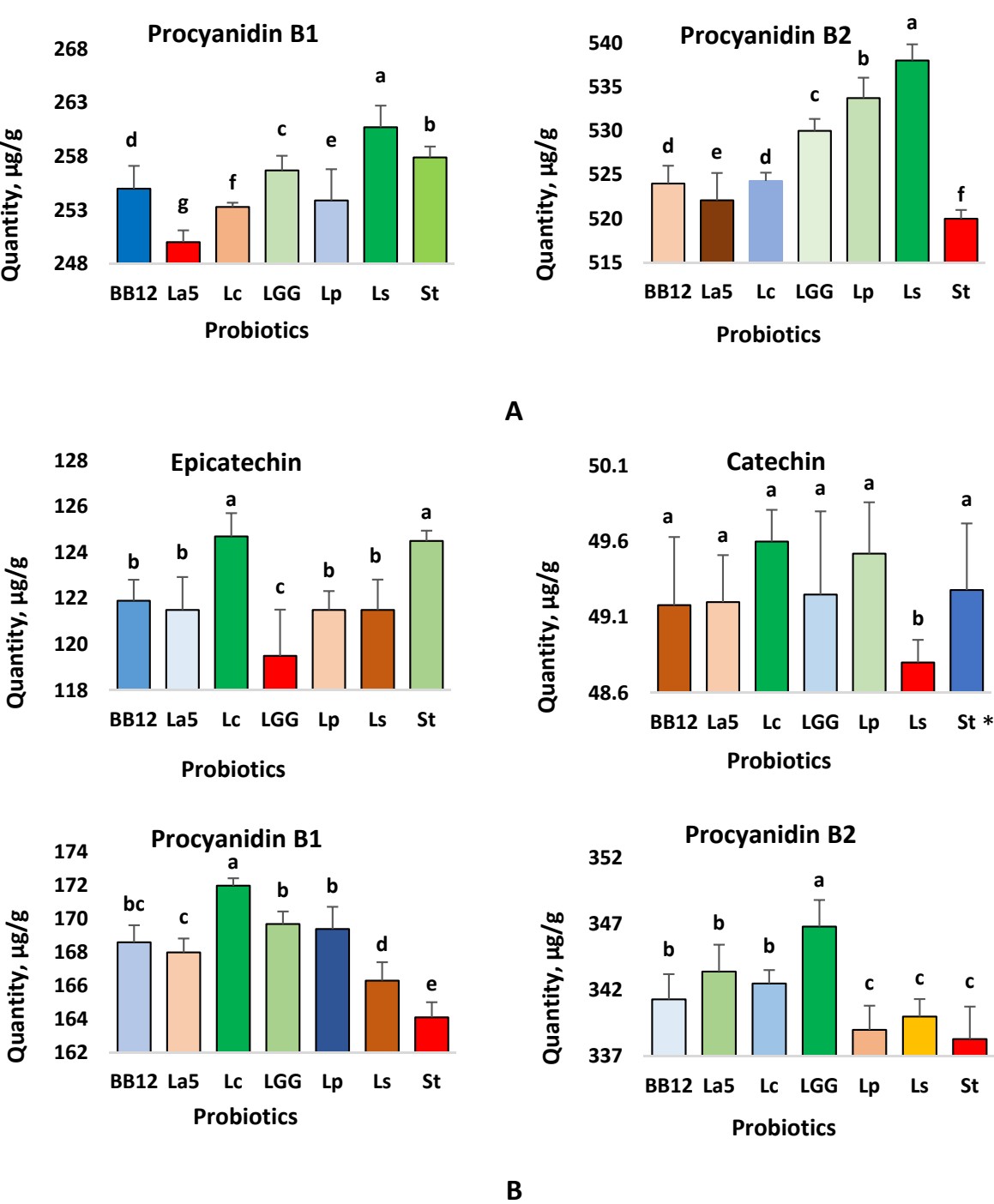

**Figure 2.** Interactive effect of major polyphenols in 70% (**A**) and 45% (**B**) cocoa chocolates through simulated in vitro gastrointestinal digestion and colonic fermentation. * BB12: *B. animalis* spp. *lactis*, La5: *L. acidophilus*, Lc: *L. casei*, LGG: *L. rhamnosus*, Lp: *L. plantarum*, Ls: *L. sanfranciscensis*, St: *S. thermophilus*. Column with different lowercase letters for each polyphenol indicate statistically significant ($p < 0.01$), $n = 6$.

## 4. Discussion

Until recently, there has not been sufficient work conducted to investigate the interactions between individual probiotics and chocolate polyphenols through in vitro gastrointestinal digestion and colonic fermentation, particularly the bioaccessibility and bioconversion of polyphenols. In this present study, seven diverse but common probiotics with diverse types of chocolates were investigated. The significantly ($p < 0.05$) larger contents of

polyphenols in chocolate with 70% cocoa powder in comparison with the 45% cocoa content could be attributed to the larger contents in the chocolate [29,30]. The TPCs recorded in this study were smaller than the value previously reported by Todorovic et al. [31], who stated that TPCs in the Serbian dark chocolate ranged from 7.21 to 12.65 mg GAE/g. On the contrary, our recorded TPCs in the positive control were much larger than the value (0.022 mg GAE/g) mentioned by Urbanska et al. and Gomez-Juaristi et al. [32,33] in some cocoa products, such as conventional and flavanol-rich soluble cocoa products. Such variations may be attributed to the diverse sources of cocoa powder, as well as processing and analysis methods of roasted and unroasted cocoa beans.

There were significant differences ($p < 0.05$) among all the tested probiotic species within each treatment (Figure 1). The negative controls (fecal slurry and basal medium only, no added probiotics) did not show any detectable amounts of polyphenols; consequently, no data were reported. These findings indicated that polyphenol bioaccessibility was related to the presence of probiotics in the chocolate samples during the in vitro gastrointestinal digestion and colonic fermentation. Similar observations were reported by Barisic et al. and Montagnana et al. [29,34] who mentioned that the available total polyphenol content varied due to the various bacterial culture. These remarks might indicate that as more polyphenols were bioaccessible during the in vitro gastrointestinal digestion, fewer amounts remained in the residue that was passed into colonic fermentation, yielding smaller amounts of bioaccessible polyphenols during colonic fermentation. In both types of fortified chocolates (45% and 70% cocoa contents), the polyphenols with higher bioaccessibility were detected after colonic fermentation in the presence of *S. thermophilus* and *L. sanfranciscensis* with the CA1 formulation. Furthermore, significant differences ($p < 0.05$) were detected between the two encapsulation formulations and CA1 showed a higher bioconversion effect than the CA2 formulation. These general trends in polyphenol bioaccessibility were also recorded in probiotic-chocolate containing 45% cocoa powder (Figure 1C) with both formulations (CA1 and CA2). Dala-Paula et al. [20] explained that dark chocolate (70% cocoa) contains a lot of proteins, amino acids, and polyphenols, which can be converted into other bioactive compounds through in vitro gastrointestinal digestion. The same authors [20] also reported that fortifying chocolate with more than one encapsulated probiotic strain had a positive impact on polyphenol bioaccessibility. They further indicated a positive correlation between the number of added probiotics that have better polyphenol bioconversion capacities.

Todorovic et al. [31] stated that 70% of dark chocolates contain 0.229 mg/g EC and 0.151 mg/g catechin, which supported the present study results of EC and catechin content in the 70% cocoa chocolates. Similar trends were detected in all the tested probiotics and within each individually analyzed flavonoid. These trends demonstrated direct positive relationships between the detected amounts before digestion and those in in vitro gastrointestinal digestion and colonic fermentation. The larger the amount of flavonoids in the undigested chocolate sample, the larger the quantity detected after in vitro gastrointestinal digestion and colonic fermentation. Similar observations were reported by Cantele et al. and Gonzalez-Barrio et al. [35,36], who reported that bioaccessible flavonoids are available at both phases of gastrointestinal digestion (in vitro gastrointestinal digestion and colonic fermentation), which were affected by the initial amount before digestion. The current results were matched with other findings, where QC gal and QC glu availability were reported at the in vitro gastrointestinal digestion stages in dark chocolates [4,35,37]. The detected quantities of QC gal and QC glu in PCh were larger than EC, C, and Pro B1, and less than Pro B2 (Table 1). Finally, our results showed that the amount of each detected flavonoid in PCh before digestion showed no significant ($p > 0.05$) differences among all the tested probiotics. Furthermore, the reported amount of each flavonoid after in vitro gastrointestinal digestion and within each tested probiotic was significantly ($p < 0.05$) higher than the quantity detected after colonic fermentation within the same treatment. The data in Figure 2 indicated that the examined probiotics showed significant differences ($p < 0.05$) in their bioconversion capacity. For example, in the chocolate with 70% cocoa, Ls and St had the highest bioconversion capacities among all the probiotics and produced

more bioaccessible flavonoids, such as EC, Pro B1 and ProB2. On the other hand, in the chocolate with 45% cocoa, Lc, LGG and Lp had significantly ($p < 0.05$) higher conversion capacities of EC, C, Pro B1 and Pro B2 flavonoids than other probiotics.

Moreover, the amount of each identified flavonoid in PCh with 45% cocoa was smaller than the amount reported with 70% cocoa. Previous studies by Michael et al. and Gottumukkala et al. [37,38] also mentioned the presence of flavonol compounds, such as epicatechin, catechin, procyanidin B1 and procyanidin B2 in cocoa and cocoa derivative products, such as chocolate products. Todorovic et al. [31] stated that 70% of dark chocolates contained 0.23 mg/g epicatechin and 0.15 mg/g catechin, which are very similar to the amounts reported in this study. The previous polyphenols have been reported to alter the gut microbiota and specifically increase the abundance of health-promoting Lactobacillus and Bifidobacterium, and decrease the amount of pathogens, without affecting other beneficial microbes [33,39]. A likely mechanism is that the high content of polyphenols was able to reduce the oxidative stress that occurs within the gastrointestinal environment, which otherwise would have caused probiotic death. This suggests that the fortification of chocolate with probiotics is a highly effective way to improve its functionality, as mediated by changes in the gut microbiota upon consumption [18,40,41]. This was also indicated in another study by Faccinetto-Beltran [42], who reported that the addition of probiotics to chocolates improved the bioaccessibility and bioavailability of bioactive compounds through animal models or clinical studies. The present study added additional value by assessing the interaction of probiotics and chocolate polyphenols to enhance the bioaccessibility during in vitro gastrointestinal digestion.

## 5. Conclusions

In this study, probiotic-chocolate was successfully produced using 1% encapsulated probiotics. It was found that the interactions between chocolate polyphenols and encapsulated probiotics can impact their bioaccessibility during in vitro gastric digestion and colonic fermentation. It was also found that the number of bioaccessible polyphenols was dependent on the metabolic activity of these probiotics. In vitro gastrointestinal digestion and colonic fermentation of PCh revealed that most bioaccessible polyphenols in both types of PCh with 45% and 70% cocoa were released in the in vitro gastrointestinal digested supernatant. The bioaccessible polyphenols from the PCh with 70% cocoa reached 83.22–92.33% during in vitro gastric digestion and 8.08–15.14% in colonic fermentation. The results also showed that the polyphenols with higher bioaccessibility during colonic fermentation of probiotic-chocolates with 45% and 70% cocoa and CA1 formulation were detected in the presence of *S. thermophilus* and *L. sanfranciscensis* probiotics. It was also noted that PCh fortified with specific probiotics would favor the production of a specific flavonoid. For example, *L. casei* released larger quantities of epicatechin and procyanidin B1, while *L. plantarum* produced more catechin, and *L. rhamnosus* LGG more procyanidin B1. The findings of this study also revealed that chocolates with a higher amount of cocoa powder (70% cocoa chocolate) may have a prebiotic impact on the growth of gut microbiota in the gastrointestinal environment. Based on the study results, it can be concluded that the most common and diverse probiotics might have the capacities to convert bioaccessible bioactive compounds in the gastrointestinal transit. The findings will be helpful to enhance the functionality of chocolates fortified with probiotics. To confirm the present findings, more in vitro and in vivo laboratory-based research needs to be conducted with other types of chocolates.

**Author Contributions:** The main project conceptualization was carried out by M.N.H. and S.A. Methodology, formal analysis, investigation, data curation and original draft writing by M.N.H. Supervision, review, editing or written materials, and instructions by S.A., C.S.R. and Z.F. All authors have read and agreed to the published version of the manuscript.

**Funding:** This work was funded by the Bangabandhu Science and Technology Fellowship' 2018, Bangabandhu Science and Technology Fellowship Trust, Ministry of Science and Technology, People's Republic of Bangladesh for financial support to Md Nur Hossain for his PhD studies.

**Institutional Review Board Statement:** An ethical approval was taken from the Human Ethics Advisory Group, The University of Melbourne, Australia. Approval ID: 1954660.1.

**Informed Consent Statement:** Informed consent was obtained from all subjects involved in the study and written informed consent has been obtained from donner to publish this paper.

**Data Availability Statement:** All the data and materials are available in this manuscript.

**Acknowledgments:** The authors want to thank Pangzhen Zhang at the School of Agriculture and Food, The University of Melbourne for his kind permission to use the polyphenol standards.

**Conflicts of Interest:** The authors have no conflict of interest to declare.

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
