# Peer review of "Interaction between Chocolate Polyphenols and Encapsulated Probiotics during In Vitro Digestion and Colonic Fermentation"

_fermentation, doi:10.3390/fermentation8060253_

Round 1
Reviewer 1 Report
The paper "Interaction between chocolate polyphenols and encapsulated 2 probiotics during in-vitro digestion and colonic fermentation" by Hossain et al. investigated the interaction between probiotics and polyphenols in chocolates (45% and 70% cocoa) enriched with encapsulated probiotics. The work quality is suitable in general, however, the following points should be addressed:
- Please have a closer look at the author’s guidelines in the case of format corrections
- First define an abbreviation then use it for the whole text, e.g. L. casei
- I would suggest the following literature to have a better idea about the polyphenols interactions and chocolate as a probiotic carrier:
- Konar, N., et al. (2022). Improving Functionality of Chocolate. Trends in Sustainable Chocolate Production, 75-112.
- Rashidinejad, A., et al. (2021). Addition of milk to coffee beverages; the effect on functional, nutritional, and sensorial properties. Critical Reviews in Food Science and Nutrition, 1-21.
- Faccinetto-Beltrán, P., et al. (2021). Chocolate as Carrier to Deliver Bioactive Ingredients: Current Advances and Future Perspectives. Foods, 10(9), 2065.
- The Folin–Ciocalteau or Lowry method has some limitations due to the presence of a number of interfering substances; including ammonium sulfate, thiol reagents, sucrose, EDTA, Tris, and Triton X-100. How do you justify it for the chocolate and also digesta with loads of interfering agents?
- A better quality (visualization) of the presented figures is needed, the colored pictures could show the work outputs in a better way, and also some figures seem to be stretched!!
- The HPLC chromatograms are not necessary here, you can move them to the supplementary material
Reviewer 2 Report
- -L57-58, rename the used strains according to the use the new taxonomy proposed from April 2020 and check all thought the manuscript
- -L102-103 correct the format of numbers (superscript or subscript)
- -L102-164, clarify the sentence as the meaning is incomplete
- -L173 correct (after before)
- -correct the formatting of all scientific names through the Ms ..see for example L207,208,211,219,220,224,234,260, 280, 283, 312,315,338,342,344,348,349,350,353,356,……etc
- Correct the formatting of all scientific names through all tables and figure legend L243-244, L303-304, L330-331
- 2 Add arrows to clarify the peaks with respective numbers, clarify the formulation, and treatment used for these data
- Table 1 and 2 Add full names for probiotics used in columns and also full names for phenolic compounds in rows
- L234 correct (on average)
- L275 correct +(-)
- Section 3.3 Figure 3B is missing in the paper (add this figure)
- The article is missing for deep discussion of all provided results, no discussion about the role of bacteria, encapsulation process, metabolic activities, and importance of the obtained results
- All figures and tables should be provided with the data of control to clarify the differences and the effect of treatments
- The authors should clarify the novelty, final recommendations and best results for possible applications
Reviewer 3 Report
The article is of relevance. But the writing of the document requires a thorough revision as there are major inconsistencies that make the manuscript deficient.
Minor issues- typos
- Line 18: CA1 is not described in the abstract
- Line 49: exclude from the parenthesis “and non-flavonoid compounds” to make it easier to read
- Line 57: the word probiotics is missing in “common (Lactobacillus rhamnosus…”
- Line 83: CA2 do not contain alginate? Improve readability to make it clear
- Line 85: odd chose of words: “resuscitated”
- Line 91-93: this sentence could be after the chocolate preparations to improve legibility
- Line 103: concentration-s S missing
- Line 117: state the sonication conditions
- Inconsistent format of stating the location of headquarters. Homogenize if the state or city is used additionally to the country (lines 88-96, 118)
- Line 137: polyphenolics determined by HPLC were only flavonoids, consider using the word “flavonoids” instead of phenolic compounds
- Line 170: the abbreviation TPC should be used in all the manuscript starting from the methodology
- Line 171: “nonencapsulated” space missing
- Line 173: “… after before and after gastrointestinal…” improve redaction
- Line 178: “contents as cocoa powder in the chocolate” improve redaction
- Line 207 to 234-243-244-260: italics missing in the scientific names
- Line 236: “biotics has some impact on polyphenols’ bioaccessibility” state the impact in some of numerical form
- Line 281-284: “2.19 μg/g in undigested, after in-vitro” undigested sample? Improve redaction
- Line 286-288: “quercetion”
- Line 311: “The highest bioaccessible epicate…” improve redaction
Inconsistencies:
- Inconsistent use of &
- in vitro is not written in italics: Lines 59-97-166-175-197-199-208-211-226-230-236-252-256-261-263-269-271-269-271-274-277-281-287-292-318-342-361
- Line 99: use the abbreviations stated in the abstract for in-vitro gastrointestinal and colonic fermentation model, or remove them from the abstract as they are not used in the manuscript nor the figures
- Line 168-170: inconsistent use of “total polyphenol”- “total polyphenols”
- Inconsistent use of the terms “phenolic compounds”, “polyphenols”, “polyphenol compounds”
- Inconsistent “probiotic-chocolates” and “probiotic chocolates”
- Inconsistent use of the (PCh) abbreviation across all the manuscript
- If there is going to be abbreviations used for the flavonoids quantified, please use the abbreviations across all the manuscript
- Inconsistent description of the samples with: “70% and 45% cocoa powder" and “samples containing 70% and 45% cocoa powder”. The latter is more clear
- Line 289: the term “Pro B1” was not stated before as an abbreviation. Use the abbreviations across all the manuscript
- Inconsistent use of “epicatechin (EC)” abbreviation
- Inconsistent use of isomer notation of catechin “(+)-catechin” in line 34 and “+(-) catechin” in line 275
- Inconsistent used of TPCs and TPC
- Inconsistent number of decimals used: “4.512±0.18 and 2.813±0.05 mg” and “87.34-92.33% and 88.72-95.61%”
- In consisted use of “enriched chocolates” and “fortified chocolates”
- Inconsistent use of “nonencapsulated probiotics” and “free probiotics”
Mayor issues
Introduction:
- Define bioavailability and bioaccessibility, as the terms are used across all the manuscript but there is no definition across
- Line 53: reference 17 does not seem as relevant as the other ones used to exemplify the microbiota fermentation by-products
Methods:
- Line 67-146: state the isomers of the reagents used
- Line 82: the FOS used are not listed in chemicals and reagents, state the brand of the product and the degree of polymerization
- Line 111-112: state if there was triplicate repetition on each sample time, or if the sample collection was from the same experimental unit.
- Feces donor: in line 105: (32yr old) a space missing. State the morphometric conditions of the “healthy male donor” and the criterion used to describe the donor as healthy. There is no information regarding ethical approval and/or consent given by the feces donor
- It is not clear in the methodology if the enrichment was a mix of the probiotics or if cholates were made with each probiotic
- State the abbreviation used for all the probiotic strains, because in results and figures are used
- Line 83: CA2 do not contain alginate? Improve readability to make it clear
- State the experimental unit of fermentation: vessels used, how anaerobic conditions were maintained, how the sample collection was made
- A table with the formulations and codification of samples could be useful to solve all the inconsistencies of notation and confusions
- The was not a negative control of the probiotic enrichment? As it would be very insightful to know it the presence of the probiotics increases the bioaccessibility and bioavailability of phenolic compounds in these food matrices
Figures:
- The figures´ quality is low, the letters are distorted widthwise, and are very pixelated
- Remove the outline of the figures
- The axis of the figures should be in the same format as the text of the manuscript: mg GAE/g
- The figures have “unencapsulated” and the text have “nonencapsulated”
- Remove the parenthesis on “formulation (CA1) –(CA2)”
- In figure 3, it is no clear what data is presented. The total contents or the released quantity in a specific phase?
- There is no need of show the chromatograms, as the quality of the images are poor (the axis is cropped, the numbers cover the lines, the lines are pixelated…) and the information is summarized in a table.
- Table 1 is very cramped, use the abbreviations stated for the digestion states in the manuscript to make some room for the numbers. Or, at discretion of the authors, put the table in supplementary material and elaborate a figure for each flavonoid quantified during all the simulated process. Could be insightful to illustrate the changes of each flavonoid in the same form as the figure 1
Discussion
- There is no discussion regarding the different formulations and the FOS content as it is an important parameter and the FOS effects on microbiota are very well established and reported
- Discussion section could be extended as it can be more in-depth about the differences withing each prebiotic strain used, and limitations of the research and future research.
- Discussion: in line 226 the sentence “These findings indicated that polyphenols bioaccessibility were directly related to the probiotic number present during the in-vitro gastrointestinal digestion and colonic fermentation” but in the manuscript there is no information of the bacterial load of each chocolate formulation. If it is reported in other paper, please specify, and used the information accordingly in the discussion
- Expand more in the discussion about the differences found in each probiotic strains or at least in the ones with significant differences. As the discussion is focused on the quantity of phenolic compounds present in each digestion phase, but there is not much discussion regarding the different strains used and the importance of evaluate all of them, and the different mechanisms of bioconversion of flavonoids by the different probiotic strains
- Please state the limitations of the research acknowledged and future research
Conclusion
- The statement: “bioaccessible polyphenols quantity depend on their metabolic activity” is not supported by the discussion
General doubts and insights
- Not sure if “probiotic chocolates” should be only used in the chocolates with the encapsulated probiotics, as the chocolate with free probiotics could share the probiotic effect
- Change “before digestion” to “initial”
Reviewer 4 Report
Dr. Hossain and colleagues conducted this experimental study in order to evaluate the interaction between probiotics and polyphenols in chocolates (containing 45% and 70% cocoa) enriched with encapsulated probiotics. The bioavailability of flavonoids and other phenolic compounds (potentially useful for the prevention of several diseases and conditions) may be influenced by the presence of different probiotics as previously reported; thus, the authors assessed the influence of 7 different probiotics in an experimental in vitro model of gastrointestinal digestion and colonic fermentation.
The authors reported a significantly higher total polyphenol contents (TPC) in chocolates with 70% cocoa powder in comparison with the 45% cocoa content. They also found significantly less bioaccessible TPC after the colonic fermentation. Concerning the influence of different probiotic strains, the authors reported: a) higher bioaccessible polyphenols after colonic fermentation in the presence of S. thermophilus and L. sanfranciscensis with CA1 formulation; b) chocolates containing 70% cocoa powder and CA1 formulation exhibited significantly larger amounts of bioaccessible polyphenols in the presence of L. plantarum after gastrointestinal digestion; c) similar results for probiotic-chocolate containing 45% cocoa powder; d) L. sanfranciscensis, S. thermophilus (and to a minor extent BB12, L. acidophilus, and L. plantarum) had better bioconversion effect on most identified phenolic compounds than other probiotics except catechin.
The study is interesting, but there are some comments that need to be addressed:
- Introduction, line 57: the word "probiotics" is missing before the bracket;
- Methods, line 84: "...were found to be..."
- Lines 114-115: validity (i.e. quality control and so on) of market products? implications?
- lines 162-164: this sentence is unclear; please rephrase.
- Figure 1 and 2: significant differences between groups are not clearly understandable; please fix;
- Tables 1 and 2 are too busy. Could the authors simplify them in order to be more immediate and informative?
- the authors did not state (and should provide discussion about) study limitations, speculations about the advantage of using such a technique for human health, implications for future research and food preparation/clinical testing;
- The choice to combine results and discussion has to be approved by the editor;
- Overall, there are many typos and minor grammatical errors that need to be fixed.
Round 2
Reviewer 2 Report
The comments have been addressed to great extent.
Author Response
English language and style are checked and edited carefully and thoroughly.
Reviewer 3 Report
After reviewing the document once again, I find it adequate. I found 2 finger bugs. Line 89 should say probiotics. Line 393 should say studies
Author Response
Thanks to the reviewer. Lines 89 and 393 have been revised accordingly.